# Preclinical Assessment in Transgenic NOD Mice of a Novel Immunotherapy for Type 1 Diabetes: Lipoplexes Down-Modulate the Murine C1858T *Ptpn22* Variant In Vitro

**DOI:** 10.3390/ijms262311241

**Published:** 2025-11-21

**Authors:** Irene Mezzani, Antonella Accardo, Emanuele Bellacchio, Luca Fais, Carlo Diaferia, Alessandra Fierabracci

**Affiliations:** 1Bambino Gesù Children’s Hospital, Scientific Institute for Research, Hospitalization and Healthcare (IRCCS), 00146 Rome, Italy; irene.mezzani@opbg.net (I.M.); luca.fais@opbg.net (L.F.); 2Research Centre on Bioactive Peptides (CIRPeB), Department of Pharmacy, University of Naples Federico II, 80134 Naples, Italy; antonella.accardo@unina.it (A.A.); carlo.diaferia@unina.it (C.D.); 3Molecular Genetics and Functional Genomics, Bambino Gesù Children’s Hospital, Scientific Institute for Research, Hospitalization and Healthcare (IRCCS), 00146 Rome, Italy; emanuele.bellacchio@opbg.net

**Keywords:** *Ptpn22*-R619W variant, gene silencing, lipoplex delivery system, Type1 diabetes (T1D), immunotherapy, NOD transgenic mice

## Abstract

The C1858T *PTPN22* (R620W) variant has been implicated in the pathogenesis of several autoimmune disorders and represents a promising immunotherapeutic target for Type 1 diabetes. We have been implementing a novel immunotherapeutic approach based on the use of lipoplexes that deliver siRNA duplexes. The efficacy and safety of lipoplexes was previously demonstrated in vitro in halting variant expression in the peripheral blood of patients. Preclinical safety and efficacy must be ascertained in vivo in appropriate animal models before clinical investigations can be undertaken, according to regulatory authorities in Europe. In the light of the foregoing, this study aims to verify that lipoplexes against the murine *Ptpn22*-R619W, equivalent to the human *PTPN22*-R620W, could be used for animal experimentation. The murine fibroblast cell line L929 was transfected with the PF62-pLenti*Ptpn22*-R619W plasmid. We designed specific siRNA duplexes for the *Ptpn22*-R619W allele and formulated them into cationic lipoplexes in order to halt variant expression in the transfected L929 cell line. Transfection of fibroblasts expressing R619W using lipoplexes resulted in efficient silencing at 100 pmol siRNA after 48 h post-transfection, reaching higher significant knockdown after 72 h. Lipoplexes efficiently suppress pathogenic *Ptpn22* variant expression in vitro, supporting the feasibility of a pre-clinical platform for testing of in vivo lipoplexes in CRISPR-engineered NOD/ShiLtJ mice carrying the R619W mutation.

## 1. Introduction

Type 1 diabetes mellitus (T1D) and autoimmune thyroid disease (ATD) frequently coexist as autoimmune polyglandular syndrome type 3 variant (APS3v), sharing a pathogenic basis in the destruction of endocrine cells by autoreactive T-cells [1,2]. The worldwide incidence of T1D is rising, especially in children <5 years, and is often accompanied by ATD, underscoring the need for therapies that preserve residual β-cells and thyrocytes [3]. The substitutive administration of the deficient hormones, namely insulin and levo-thyroxine (L-T4), is the standard treatment, although this does not halt the autoimmune process and does not rescue the residual hormone-producing cells [4]. Identification of innovative therapeutic interventions, especially those aimed at preserving the residual cells, is of crucial importance in improving the expectation of quality of life in pediatric patients [5].

Recently, the potential pathophysiological role played by the *PTPN22* C1858T mutation in several autoimmune disorders, including T1D and APS3v, has been demonstrated [6,7,8,9,10]. *PTPN22* encodes the protein tyrosine-protein phosphatase non-receptor type 22 (also referred to as lymphoid phosphatase, Lyp), which is a T-cell antigen receptor (TCR) signaling negative regulator, acting in concert with C-Src kinase. In Lyp, the C1858T variant produces the replacement of arginine 620 with a tryptophan residue (R620W), which leads to a gain of function with paradoxical reduced T-cell activation, affecting both innate and adaptive immune responses [11,12,13,14].

In light of this fact, it may be a valid drug target for the treatment of T1D and APS3v patients. We therefore implemented a novel immunotherapy based on the use of lipoplexes delivering siRNA that cause variant-allele-selective inhibition instead of complete gene knockdown [15,16,17,18,19]. We have already demonstrated the efficacy and tolerability of lipoplexes in peripheral blood mononuclear cells (PBMC) of T1D patients harboring the variant *PTPN22*. Furthermore, we developed strategies for functionalization of lipoplexes to improve selective delivery to specific immunocytes, in particular, and to allow the possibility of exposing Fab of monoclonal antibodies [19] to achieve specific silencing in B lymphocytes and cytotoxic T cells. An alternative functionalization strategy to target several immunocytes in the peripheral blood was exploited using a high-affinity Siglec 10 sialoside mimetic (SAM, PEG-lipid-F9) [19,20,21]. The Siglec family of sialic-acid-binding proteins is primarily expressed on cells of the immune system, which mediate innate and adaptive responses [21] functioning in the self and non-self discrimination. Thus, these endocytic receptors became attractive immunotherapeutic targets [21]. We proved the internalization and low toxicity of Lipo-siRNAR620W-PEGF9 (LiposiRNA-PEGF9) in PBMC and their efficacy in halting variant mRNA expression, as well as their functional efficacy by rescuing IL-2 secretion in PBMC of C1858T *PTPN22* T1D patients. Lipoplexes were functionalized with sialoside mimetic PEGF9 and optimized for in vivo preclinical delivery studies [22].

In view of the prospective use of lipoplexes as immunotherapy of T1D and APS3v, a fundamental step forward is to conduct preclinical studies in animal models. For this purpose, according to international regulations, the biodistribution of lipoplexes and their functionalized compounds must be assessed by injection of fluorescently labeled compounds in murine strains such as C57BL/6 or BALB/c mice, commonly used for this kind of evaluation [23] as well as safety/efficacy in delaying or blocking the development of the disease. The NOD/ShiLtJ mouse (non-obese diabetic mice), a genetically predisposed strain for the development of autoimmune diabetes, closely mimicking the human model, appears to be ideal for preclinical studies to further investigate the influence of the R620W gene variant that causes the human disease, and to evaluate the efficacy of the novel pharmacological treatment [24,25,26]. In particular, phenotyping of the NOD model confirmed a 90% incidence of diabetic females by 30 weeks of age. The ideal NOD model carrying the R619W-*Ptpn22* variant, equivalent to the human R620W, was generated using CRISPR-Cas9 technology [27]. This transgenic animal exhibits higher levels of anti-insulin antibodies, an earlier onset of the disease, and a higher prevalence in females.

In unraveling the feasibility of undertaking the proposed line of experimentation to ascertain the efficacy and safety of lipoplexes in the NOD transgenic mouse, we first aimed to effectively demonstrate the possibility of inhibiting the murine variant in vitro by using lipoplexes carrying the respective siRNA duplexes in the R619W-transfected L929 murine fibroblast cell line.

## 2. Results

### 2.1. Transfection of the R619W-Ptpn22 Variant in the L929 Mouse Fibroblast Cell Line

The mouse L929 cell line (ATCC^®^ CCL-1™, American Type Culture Collection, Manassas, VA, USA) was transfected with the PF62-pLenti*Ptpn22*-R619W plasmid (Aurogene S.r.l., Rome, Italy).

The quantitative Real-Time PCR (qRT-PCR) results of transfection with the PF62-pLenti*Ptpn22*-R619W plasmid versus empty vector (EV) are shown in the table below (Table 1).

### 2.2. siRNAs Are Delivered in the R619W-L929 Cell Line by a Commercial Transfection System and Efficiently Block Variant Ptpn22 Expression

In the experiment performed using the transfection kit to convey a siRNA sense/antisense (s/a) duplex with Lipofectamine (Lipofectamine™ RNAiMAX, Invitrogen, Thermo Fisher Scientific, Waltham, MA, USA), which has a higher affinity for the target mRNA sequence, after 48 h of transfection, we observed a maximum inhibition of R619W mRNA in the range between 10 and 80 pmols of siRNA compared to untreated control cells. After 72 h from the beginning of the transfection, a higher degree of inhibition persisted at a 40 pmol concentration of siRNA duplex (Figure 1A,B).

These results highlight the specificity of the siRNA duplex, selected for the subsequent experiments of lipoplex variant inhibition reported below (vide infra).

### 2.3. Efficient Silencing of Variant Ptpn22 in R619W-L929 Cells by siRNA Delivery Using Lipoplexes

Lipoplexes efficiently block variant *Ptpn22* expression in R619W-transfected L929 cells.

Within the range of siRNA doses between 60 and 100 pmols, the highest inhibition of variant mRNA was obtained with 100 pmol of siRNA after 48 h of transfection (Figure 2A). The efficiency increased after 72 h from the beginning of transfection and was shown even at the lower doses of 60 and 80 pmols of siRNA (Figure 2B).

## 3. Discussion

According to regulatory authorities in Europe (EMA, European Medicines Agency) and the US (FDA, Food and Drug Administration), the interpretation of results from animal studies is necessary to support the future clinical application of novel therapeutic modalities. Indeed, preclinical safety and efficacy assessment of new medicinal products ensures that benefits will outweigh the risks associated with their use [23].

Nevertheless, the main challenge in developing a proof of concept for advancing to a clinical trial is, in most circumstances, the unavailability of appropriate animal models with similar functional characteristics to the intended human disorder.

A current line of investigation in our laboratory is to exploit the feasibility of the novel immunotherapy for T1D and APS3v based on the use of lipoplexes targeting the C1858T *PTPN22* variant [16,17,18,19]. In light of the arguments raised (vide supra), preclinical assessment is requested at this stage of the investigation to ascertain the safety and efficacy of lipoplexes in the ideal NOD mice model made transgenic for the variant R619W-*Ptpn22* through CRISPR-Cas9 technology [27]. For this purpose, it was necessary to have confirmation in preliminary settings of the possibility of blocking the variant R619W of *Ptpn22* mRNA in the transgenic L929 fibroblast cell line using lipoplexes carrying the appropriate siRNA duplex. The murine line was transfected with the PF62-pLenti*Ptpn22*-R619W plasmid vector (which is commercially available). Lipoplexes carrying siRNA duplexes against the variant efficiently reduced variant R619W-*Ptpn22* expression, as revealed using qRT-PCR, although with less efficacy than with the standard Lipofectamine system used as a control. Indeed, lipoplexes produced a significant inhibitory effect on variant mRNA expression at higher doses (80–100 pmols of siRNA concentration) and with a longer time interval for internalization than with the Lipofectamine transfection control. The lower silencing efficiency observed with lipoplexes compared to Lipofectamine can be determined by several factors, such as cellular uptake and intracellular processing mechanisms. Lipofectamine-based systems are optimized commercial reagents that promote efficient siRNA complexation, membrane fusion, and endosomal escape, resulting in higher cytoplasmic release of siRNA molecules. In contrast, lipoplex formulations may exhibit lower stability and less efficient endosomal release. Moreover, differences in lipid composition and charge ratio can influence the kinetics of complex disassembly and siRNA release within the cytoplasm [28].

These results encourage us to proceed with assessment through in vivo animal experimentation on lipoplexes, the results of which will possibly allow future advancement to Phase I/II clinical trials of T1D immunotherapy in adults, which are necessary before a pediatric investigation plan can be implemented.

## 4. Materials and Methods

### 4.1. siRNA Design

Authentic siRNA sequences were designed (Design ID: ABHSP52) to specifically target the C1858T *Ptpn22* murine gene variant (Life Technologies, Thermo Fisher, Milan, Italy).

### 4.2. Liposome/Lipoplex Preparation

Phospholipids [1,2-Dioleoyl-sn-glycero-3-phosphoethanolamine (DOPE), N-[1-(2,3-Dioleoyloxy) propyl]-N,N,N-trimethylammonium chloride (DOTAP), 1,2-distearoyl-sn-glycero-3-phosphoethanolamine-N-[maleimide (polyethylene glycol)-2000] (ammonium salt) (DSPE-PEG2000-Maleimide)] were acquired from Avanti Polar Lipids (Alabaster, Alabama). Liposomal solutions were assembled in 100 mM phosphate buffer (PBS, Euroclone, Milan, Italy) at a pH of 7.4. DOTAP/DOPE/DSPEPEG2000-Maleimeide (47.5/47.5/5 molar ratio) liposomes were formulated using the lipid film method as previously described [18]. Briefly, phospholipids previously dissolved in chloroform were mixed, and the organic solvent was evaporated under a nitrogen stream to obtain a homogeneous film. The film was subsequently hydrated with 1.0 mL of 100 mM PBS (pH = 7.4) to achieve a final concentration of 800 μM. The suspension was sonicated for 30 min (min) and successively extruded 10 times at room temperature (RT), using a thermo barrel extruder system (Northern Lipids Inc., Vancouver, BC, Canada) under nitrogen through a polycarbonate membrane (Nucleopore Track Membrane, 25 mm, Whatman, Brentford, UK) with a pore size of 0.1 μm. Then, 500 μL of a siRNA solution (5.2 μM in water) was mixed with an equivalent volume of liposomes, and the resulting solution was stirred at RT for 3 h.

Lipoplexes were generated by mixing an equal volume of the liposomal solution and 2.6 μM siRNA at RT for 3 h. The resulting solution contains lipoplexes in which both the liposomal and siRNA concentrations are halved. The mean diameter of lipoplexes (150 nm) did not show differences with respect to the formulation previously prepared by us using the same liposomal composition and the human variant of the siRNA [19] (Figure 3).

### 4.3. Generation of R619W-Transfected L929 Murine Fibroblast Cell Line

The L929 mouse line was transfected with the R619W mouse variant using the PF62-pLenti*Ptpn22*-R619W plasmid available from Aurogene S.r.l. (Rome, Italy).

### 4.4. L929 Cell Culture

R619W-*Ptpn22*-transfected L929 cells (ATCC^®^ CCL-1™, American Type Culture Collection, Manassas, VA, USA) were cultured at a density of 2 × 10^5^ cells/mL in T75 flasks (Falcon Labware Becton Dickinson (BD) Biosciences, Oxnard, CA, USA). Cells were cultured in complete Dulbecco’s Modified Eagle Medium high-glucose medium (DMEM HG, Euroclone, Milan, Italy) supplemented with 10% fetal bovine serum (FBS) (Hyclone, South Logan, UT, USA), L-glutamine (2 mM, Euroclone, Milan, Italy), and 1% penicillin/streptomycin (pen/strep) (Euroclone), incubated at 37 °C in a 5% CO_2_ humidified atmosphere, and sub-cultured twice per week at the same seeding density.

### 4.5. Mutant Ptpn22-Gene Silencing in a Transfected L929 Fibroblast Line (R619W-L929)

#### 4.5.1. siRNA Transfection Using a Commercial Transfection System

Selected siRNA sense/antisense (s/a) duplexes (R619W-Forward: 5′-TCCCCTCCGAATAGTGCTGA-3′-R619W-Reverse: 5′-CATTCAGGGAGTGGCGG-3′) with R619W mRNA target affinity were first validated by measuring their capability to silence L929 cells using a commercial transfection kit (Lipofectamine™ RNAiMAX, Invitrogen, Thermo Fisher Scientific, Waltham, MA, USA) and observing the resulting decrease in mRNA levels through qRT-PCR.

Briefly, on the first day, L929 cells were rescued from T75 culture flasks using a trypsin/EDTA solution (Euroclone), washed in PBS (Euroclone) and seeded at 1.5 × 10^5^ per well in 12-well plates (Falcon, Corning Incorporated, Corning, NY, USA) in 1 mL of DMEM HG (Euroclone) supplemented only with 1% pen/strep and L-glutamine (2 mM) and then maintained for 24 h at 37 °C in a 5% CO_2_ humidified atmosphere.

On day 2, cells were washed with PBS (Euroclone) and then treated with different concentrations of R619W-*Ptpn22* siRNA duplex (10, 40, 80 pmols of siRNA) in transfection medium (Gibco, Thermo Fisher Scientific) and incubated overnight (O/N) at 37 °C in a 5% CO_2_ humidified atmosphere. To evaluate siRNA internalization efficiency, cells were incubated using the fluorescent (Alexa Fluor 555-conjugated) siRNA provided by the manufacturer.

On day 3, complete DMEM HG medium (Euroclone) supplemented with 20% FBS and 2% pen/strep was added to stop the O/N incubation. The control group was treated with the fluorescent siRNA and subsequently analyzed by flow cytometry using a FACS Canto II instrument (Becton and Dickinson, Sunnyvale, CA, USA) and PC FACS Diva software (BD Biosciences, San Jose, CA, USA).

On day 4 and day 5, after 48 and 72 h, respectively, from the beginning of each transfection with different concentrations of R619W-*Ptpn22* siRNA duplex, cells were harvested, washed with PBS, pelleted at 500× *g* for 5 min, and then lysed in RLT buffer (Qiagen, Hilden, Germany) for subsequent RNA extraction. Each experimental condition was assessed in two independent replicates.

#### 4.5.2. Custom Liposome Transfection Protocol of R619W-L929 Fibroblast Line

On the first day, L929 cells were harvested from T75 culture flasks using a trypsin/EDTA solution (Euroclone) and cultured at 0.7 × 10^5^ cells per well in 24-well plates (Falcon) in 500 μL of DMEM HG (Euroclone) enriched with 10% FBS, 1% pen/strep and L-glutamine (2 mM) for 24 h at 37 °C in a 5% CO_2_ humidified atmosphere.

On day 2, cells were first rinsed with PBS (Euroclone) and then cultured using 500 μL of the same FBS and pen/strep-free medium, containing different amounts (60, 80, 100 pmols) of siRNA duplex in relation to the *Ptpn22* variant (vide supra), assembled as lipoplexes. Cells were incubated O/N with the complexes at 37 °C in a 5% CO_2_ humidified atmosphere. Appropriate controls were set up by growing cells using complete DMEM HG (Euroclone) medium alone. Each experimental condition included at least three independent determinations (triplicates). At the end of the incubation period, cells were harvested from the cell culture plates, washed in PBS by centrifugation at 500× *g* for 5 min, and cultured in complete DMEM HG (Euroclone) at a final volume of 1 mL.

On day 4 and on day 5, after 48 and 72 h, respectively, from the beginning of each transfection, cells were washed in PBS and then lysed in RLT buffer (Qiagen) for subsequent RNA extraction.

#### 4.5.3. qRT-PCR

Relative gene expression levels were assessed by quantitative Real-Time PCR (qRT-PCR). Cells were lysed in RLT-Buffer at 48 and 72 h post-treatment. After RNA extraction (RNeasy Plus Mini Kit, Qiagen), quantification was performed as instructed by the manufacturer by spectrophotometry (Nanodrop 2000, Thermo Fisher Scientific, Waltham, MA, USA), followed by reverse transcription (SuperScript™ IV First-Strand Synthesis System, Thermo Fisher Scientific) and qRT-PCR amplification (PowerUp™ SYBR™ Green Master Mix, Thermo Fisher Scientific) employing the primers listed below to amplify the murine R619W-*Ptpn22* variant:R619W forward (R619W-forward): 5′-TCCCCTCCGAATAGTGCTGA-3′;R619W reverse (R619W-reverse): 5′-CATTCAGGGAGTGGCGG-3′.

Murine GAPDH was used as the internal housekeeping control. The following primers were used:GAPDH forward (GAPDH-FW): 5′-ATTCAACGGCACAGTCAAGG-3′;GAPDH reverse (GAPDH-RV): 5′-ATCATAAACATGGGGGCATC-3′.

Relative expression levels were calculated using the 2^–ΔΔCt^ method.

### 4.6. Statistical Analysis

Statistical analyses were performed using GraphPad Prism 10.0 (GraphPad Software, San Diego, CA, USA). One-way analysis of variance (ANOVA), followed by Šidák’s post hoc test for multiple comparisons, was used to identify significant differences among groups. Statistical significance was defined as a * *p* ≤ 0.05. Results are expressed as means ± standard deviation (SD) in Appendix A and % of inhibition in Appendix A.

## 5. Conclusions

Allele-specific inhibition via siRNA-containing lipoplexes represents a promising strategy for selectively silencing the *PTPN22* genetic mutation C1858T in immunocytes implicated in the pathogenesis of APS3v/T1D.

In vitro results shown in this manuscript confirm the possibility of targeting the pathogenic (R619W-*Ptpn22*) allele in CRISPR-modified NOD/ShiLtJ mice, which harbors the variant, paving the way for preclinical translation. Experimental design for safety and efficacy studies will include the use of this knocking mice model (https://www.jax.org/strain/036948 (accessed on 30 September 2025) available from Jackson Laboratories (Bar Arbor, ME, USA) [27]. Heterozygous transgenic mice, generated from two couples breeding, will be treated with lipoplexes by intravenous injection in the lateral tail vein. Animals will be monitored daily for health status assessment, weighed twice a week for the overall 7-month time course of the experimental procedure. Once a week, a blood sample will be taken from the mandibular plexus [29] to measure blood glucose. Glycosuria will be tested once a week until diabetes onset, then once a month toward the end of the experimental procedure. Lipoplexes functionalized with Fab anti-murine CD3 will also be developed and tested in the animal model in parallel experiments [18]. Biodistribution of lipoplexes will also be estimated in the Balb/c mice strain by injection, through the tail lateral vein, of lipoplexes and anti-CD3Fab functionalized lipoplexes to specifically target T lymphocytes, both labeled with ATTO647NDOPE (ATTOTEC, Siegen, Germany). Mice will be sacrificed after 48 h post-injection, and PBMC will be taken and cryopreserved. Lymphoid and non-lymphoid organs, including lymph nodes, thymus, spleen, pancreas, liver, and heart, will be snap-frozen and embedded in OCT (Optimal cutting temperature) compound. At different times within 48 h post-injection, IVIS (in vivo imaging system) will be applied to monitor lipoplexes’ biodistribution. These findings provide a solid basis for future biodistribution, pharmacokinetic, and efficacy studies as required by regulatory authorities in Europe before clinical investigations can be undertaken.

## 6. Patents

Italian Patent 102018000005182 issued on 26.05.2020; PCT/IT2019/050095 filed on 8.05.2019, extended in Europe, USA, and Hong Kong, issued in USA on 9 September 2025. Inventor: Alessandra Fierabracci, MD, PhD.

## Figures and Tables

**Figure 1 ijms-26-11241-f001:**
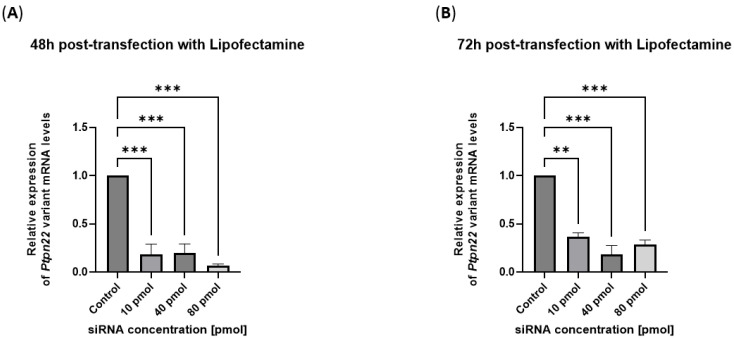
Lipofectamine-siRNA-mediated silencing of *Ptpn22* mRNA expression. (**A**) qRT-PCR analysis performed 48 h post-transfection with Lipofectamine showed a dose-dependent reduction in *Ptpn22* transcript levels, with maximum silencing at 80 pmol siRNA (0.0660 ± 0.0186; 93.4% of inhibition), compared to untreated controls. (**B**) At 72 h post-transfection with Lipofectamine, *Ptpn22* mRNA expression remained reduced across all siRNA concentrations, with the strongest silencing effect detected at 40 pmol (0.1822 ± 0.0928, 81.78% of inhibition) siRNA. Data are expressed as mean ± SEM of two replicates in Appendix A (** *p* < 0.01, *** *p* < 0.001) and as percentage (%) of inhibition in Appendix A.

**Figure 2 ijms-26-11241-f002:**
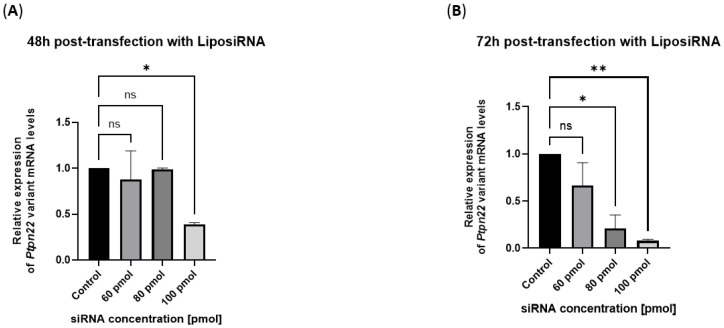
LiposiRNA-mediated silencing of *Ptpn22* mRNA expression. (**A**) qRT-PCR analysis performed 48 h post-transfection with LiposiRNA complexes showed a dose-dependent reduction in *Ptpn22* transcript levels, with maximum silencing at 100 pmol siRNA (0.3853 ± 0.0226, 61.47% of inhibition), compared to untreated controls. (**B**) At 72 h post-transfection with LiposiRNA complexes, *Ptpn22* mRNA expression remained reduced across all siRNA concentrations, with the strongest knockdown confirmed at 100 pmol (0.0788 ± 0.0150, 92.12% of inhibition) siRNA. Data are expressed as mean ± SEM of two replicates in Appendix A (ns = not significant, * *p* < 0.05, ** *p* < 0.01) and % of inhibition in Appendix A.

**Figure 3 ijms-26-11241-f003:**
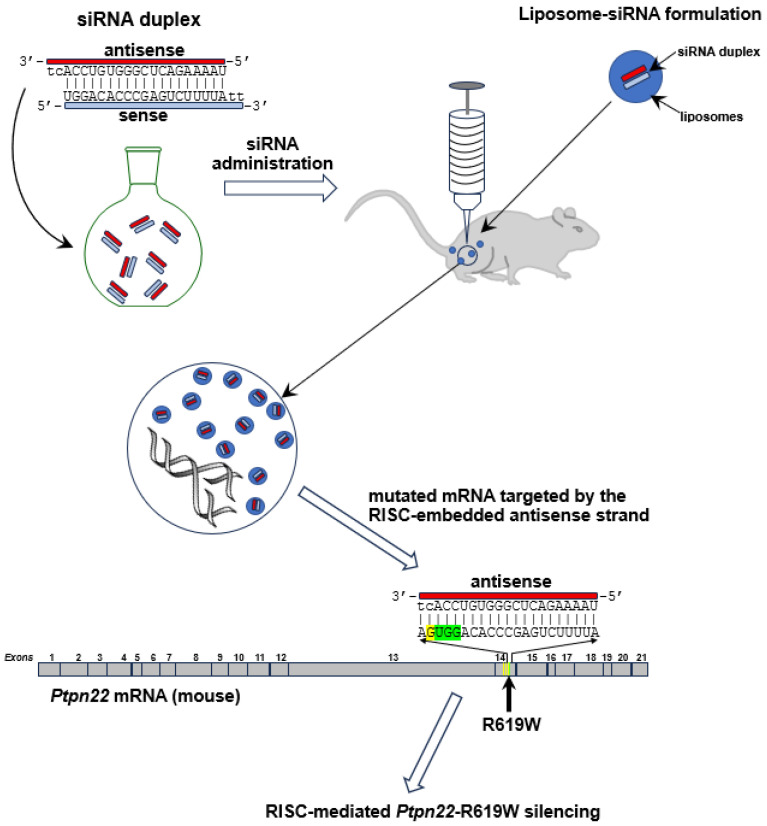
Schematic representation of the siRNA-based therapeutic strategy targeting the pathogenic R619W mutation in the mouse *Ptpn22* gene. The illustration depicts the application of a 21-nucleotide antisense siRNA strand designed to perfectly complement a segment within exon 14 of murine *Ptpn22* mRNA, specifically encompassing the mutated codon (highlighted in green). The mouse model used in this study (NOD) harbors the R619W mutation, corresponding to the pathogenic human mutation R620W, along with three additional nucleotide variants located proximally within the same exon (indicated by yellow bars on the mRNA sequence).

**Table 1 ijms-26-11241-t001:** qRT-PCR analysis. Results of L929 cells transduced with the PF62-pLenti*Ptpn22*-R619W plasmid and with the empty vector (EV) are shown. Ct, Cycle threshold.

Sample ID	*Ptpn22* Ct	β-Actin Ct	ΔCt	ΔΔCt	Fold Change (2^−ΔΔCt^)
L929-EV	36.86	16.99	−19.60	0.00	1.00
L929-EV	36.93	17.09			
L929-EV	36.98	17.35			
L929-*Ptpn22*-R619W	26.43	17.10	−10.46	−10.46	1409.15
L929-*Ptpn22*-R619W	26.21	17.14			
L929-*Ptpn22*-R619W	26.24	17.21			

## Data Availability

The original contributions presented in this study are included in the article/Appendix A. Further inquiries can be directed to the corresponding author.

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
