# Peer review of "Preclinical Assessment in Transgenic NOD Mice of a Novel Immunotherapy for Type 1 Diabetes: Lipoplexes Down-Modulate the Murine C1858T Ptpn22 Variant In Vitro"

_ijms, 2025, doi:10.3390/ijms262311241_

Round 1

Reviewer 1 Report

Comments and Suggestions for Authors

This study focuses on the novel immunotherapy for type 1 diabetes (T1D) and the variant of autoimmune polyglandular syndrome type 3 (APS3v), targeting the PTPN22 - C1858T (human R620W/mouse R619W) variant associated with the disease. Lipoplexes carrying siRNA were developed and their effectiveness in silencing mouse R619W - Ptpn22 expression was verified through in vitro experiments, laying the foundation for subsequent CRISPR editing in NOD mice in vivo experiments, demonstrating clear clinical translation potential. 

The following specific review comments are provided:
1. In the current experiment, the lipoplex transfection group only set "complete DMEM HG medium alone culture" as the control, without including the "blank liposome (without siRNA) treatment group".
2. The results section (such as 2.2 and 2.3) mentions that the data are the mean ± standard error of "two replicates" or "three replicates", but it does not clarify whether independent experiments were repeated for verification (such as at least three independent experiments).
3. The study only verified the silencing effect of lipoplexes on R619W - Ptpn22 mRNA, without further exploring their impact on downstream signaling pathways (such as TCR signaling, IL - 2 secretion).
4. Section 4.2 mentions the physicochemical characterization of lipoplexes, but does not list specific parameters (such as particle size, zeta potential, siRNA encapsulation rate, stability, etc.).
5. The conclusion mentions paving the way for CRISPR-modified NOD mice in vivo experiments, but does not describe the specific design ideas for subsequent in vivo experiments (such as administration route, dose gradient, detection indicators (such as blood glucose, islet pathology, anti-insulin antibody levels), etc.).
6. The introduction section cites many references on the association between PTPN22 variants and autoimmune diseases and the research progress of lipoplex delivery systems from 2022 and before. It is suggested to supplement the latest relevant research from 2023 to 2025 to reflect the latest trends in the field and enhance the timeliness and innovative positioning of the research.

Author Response

This study focuses on the novel immunotherapy for type 1 diabetes (T1D) and the variant of autoimmune polyglandular syndrome type 3 (APS3v), targeting the PTPN22 - C1858T (human R620W/mouse R619W) variant associated with the disease. Lipoplexes carrying siRNA were developed and their effectiveness in silencing mouse R619W - Ptpn22 expression was verified through in vitro experiments, laying the foundation for subsequent CRISPR editing in NOD mice in vivo experiments, demonstrating clear clinical translation potential. 

The following specific review comments are provided:

  1. In the current experiment, the lipoplex transfection group only set "complete DMEM HG medium alone culture" as the control, without including the "blank liposome (without siRNA) treatment group".

We thank the reviewer for this comment. The blank liposome (without siRNA) was not included as a control because empty formulations have shown some instability with cytotoxicity in preliminary tests. To avoid introducing biased effects, we used cells cultured in complete medium alone as the control, which allowed us to specifically assess the effects of the siRNA-loaded lipoplexes.

  1. The results section (such as 2.2 and 2.3) mentions that the data are the mean ± standard error of "two replicates" or "three replicates", but it does not clarify whether independent experiments were repeated for verification (such as at least three independent experiments).

For the experiments described in Sections 2.2 and 2.3, the reported data were obtained from three independent biological replicates, derived from separate cultures of the same cell line as detailed on page 10, line 343, section 4.5.2 of Materials and Methods of the revised marked version of the manuscript.
Biological replicates were then used to perform two technical replicates of Real-Time PCR (qRT-PCR) in two independent experiments, in order to verify the reproducibility of the results (4 set of data).
Accordingly, the values reported as mean ± standard error represent the analysis of data from these biological and technical replicates, confirming the robustness and consistency of the experimental results.

  1. The study only verified the silencing effect of lipoplexes on R619W - Ptpn22 mRNA, without further exploring their impact on downstream signaling pathways (such as TCR signaling, IL - 2 secretion).

For carrying on the experimentation in vitro, we elected to use the L929 fibroblast cell line, which was recommended by the company Aurogene Srl, Rome, Italy, as ideal to warrant efficiency and stability of transfection (see also reply to Reviewer 2 point 2) with PF62-pLentiPtpn22-R619W plasmid. Nevertheless, although ideal as ‘surrogate’ simple model to explore efficiency of lipoplexes in blocking the variant expression, this cell line is not however ideal for exploring the impact of lipoplexes on downstream signaling pathways (such as TCR signaling, IL-2 secretion). We believe that this could be at the best functionally evaluated in vivo on PBMC immunotypes of the animal model carrying the R619W-Ptpn22 variant. Nevertheless, we have already demonstrated in our previous publications the effects of the downstream effect on IL-2 secretion of lipolexes blocking human variant on human PBMCs (Refs. 17 and 19) of the revised marked version of the manuscript.

  1. Section 4.2 mentions the physicochemical characterization of lipoplexes, but does not list specific parameters (such as particle size, zeta potential, siRNA encapsulation rate, stability, etc.).

We thank the Reviewer for this inquire. We recently reported, in our previous study (Ref. 22 of the revised marked version of the manuscript) the structural characterization of the lipoplex prepared by incubating human siRNA with the same liposomal formulation [DOTAP/DOPE/DSPEPEG2000-Maleimeide (47.5/47.5/5 molar ratio)]. Here, we used the same liposomal formulation to deliver murine siRNA, which has the same net charge of the human variant and the same number of base pairs. As expected, we did not observe any difference between the size and more in general in the properties of the two lipoplexes differing in the siRNA variant. For this reason, we did not report structural data for lipoplex. However, we noticed that our previous sentence at page 7 row 233-234 was misleading. We changed it and explained this point more clearly in the revised version of the manuscript (see also the reply to Reviewer 2, point 4).

  1. The conclusion mentions paving the way for CRISPR-modified NOD mice in vivo experiments, but does not describe the specific design ideas for subsequent in vivo experiments (such as administration route, dose gradient, detection indicators (such as blood glucose, islet pathology, anti-insulin antibody levels), etc.).

Ideas regarding the experimental animal protocol are reported in the revised marked version of the manuscript in the Conclusions section on page 10 from line 387 of the revised marked version of the manuscript.

  1. The introduction section cites many references on the association between PTPN22 variants and autoimmune diseases and the research progress of lipoplex delivery systems from 2022 and before. It is suggested to supplement the latest relevant research from 2023 to 2025 to reflect the latest trends in the field and enhance the timeliness and innovative positioning of the research.

We updated references in the Introduction, also referring to our latest publications on the development of lipoplexes (see added references n°7, 8, 9, 10, 11, 12, 13, 14, 22, 28, 29 in the revised marked version of the manuscript).

Reviewer 2 Report

Comments and Suggestions for Authors

The study explores an allele-specific siRNA lipoplex system targeting the murine Ptpn22-R619W variant, a homolog of the human PTPN22-R620W mutation implicated in Type 1 diabetes (T1D) and autoimmune thyroid disease. The work builds on the authors’ prior studies in human PBMCs and aims to provide in vitro validation supporting as the authors state upcoming in vivo testing in CRISPR-engineered NOD mice. Overall the manuscript presents a promising concept for RNA-based immunotherapy in autoimmune diseases. However, improvements are needed considering methodological details, inclusion of appropriate controls, and expanded discussion of mechanistic aspects. Some comments follow: 

1) The rationale for allele-specific Ptpn22 silencing is clear, but the added value over the authors’ previous publications (Refs 9–12) is not sufficiently defined. How this in vitro murine model extends beyond confirming prior human PBMC results. Is the siRNA sequence or variant-specific design distinct, or is this primarily a feasibility confirmation for future in vivo testing?

2) The L929 fibroblast line is not immune-derived and does not naturally express Ptpn22.
Please justify why this line was chosen instead of a lymphoid or immune-competent murine cell line (e.g., EL-4, RAW 264.7). Discuss how this choice may limit extrapolation to immune-cell contexts.

3) The qRT-PCR data are summarized but the manuscript does not specify biological replicates, variation, or normalization details. So, numerical data (mean ± SD) and complete statistical outcomes (F, df, p-values) are needed as well as confirmation that experiments were independently repeated.

4) The text refers to “previously established protocols” but omits characterization of the actual lipoplex batch used in this study. Were zeta potential, hydrodynamic diameter, and siRNA entrapment quantified for this experiment? Please include these parameters or a supplementary figure confirming reproducibility.

5) Figures 1–2 describe dose-dependent silencing, but quantitative knock-down percentages and ICâ‚…â‚€ values are missing. Please consider to include actual silencing efficiencies at each dose and time point.

6) The absence of scrambled or non-targeting siRNA controls makes it difficult to exclude off-target or lipoplex-related effects. So, controls were included in this approach? if not, justify or include them in revised experiments.  

7) Lipofectamine. No mechanistic discussion of the observed lower efficiency is provided. Can the authors elaborate on the differences in uptake mechanism, stability, or endosomal escape that may explain the lower silencing efficacy of lipoplexes?

8) ANOVA with Šidák’s test is mentioned, yet corresponding statistics are not reported (p-values, CIs) 

9) Considering the next step to in vivo experimental approaches. What endpoints (biodistribution, immunogenicity, toxicity) are planned for the NOD model, and what are the success criteria for moving toward clinical translation?

10) There are some phrases that could be improved. Consider to proceed to a thorough grammar check to improve readability.

Comments on the Quality of English Language

There are some phrases that could be improved. Consider to proceed to a thorough grammar check to improve readability.

Author Response

The study explores an allele-specific siRNA lipoplex system targeting the murine Ptpn22-R619W variant, a homolog of the human PTPN22-R620W mutation implicated in Type 1 diabetes (T1D) and autoimmune thyroid disease. The work builds on the authors’ prior studies in human PBMCs and aims to provide in vitro validation supporting as the authors state upcoming in vivo testing in CRISPR-engineered NOD mice. Overall the manuscript presents a promising concept for RNA-based immunotherapy in autoimmune diseases. However, improvements are needed considering methodological details, inclusion of appropriate controls, and expanded discussion of mechanistic aspects. Some comments follow: 

1) The rationale for allele-specific Ptpn22 silencing is clear, but the added value over the authors’ previous publications (Refs 9–12) is not sufficiently defined. How this in vitro murine model extends beyond confirming prior human PBMC results. Is the siRNA sequence or variant-specific design distinct, or is this primarily a feasibility confirmation for future in vivo testing?

The in vitro model with the transfected L929 line was implemented in this manuscript to verify that, as in the case of the inhibition of the human variant obtained in human PBMC with an appropriate siRNA duplex, similarly, it could be possible to inhibit the murine variant with an appropriately designed siRNA duplex for R619W Ptpn22. This verification is fundamental before conducting animal experimentation of inhibition of the murine variant in vivo in the transgenic NOD mice to verify the safety and efficacy of lipoplexes delivering the same siRNA duplex appropriately designed for murine R619W Ptpn22.

2) The L929 fibroblast line is not immune-derived and does not naturally express Ptpn22. Please justify why this line was chosen instead of a lymphoid or immune-competent murine cell line (e.g., EL-4, RAW 264.7). Discuss how this choice may limit extrapolation to immune-cell contexts.

For carrying on the experimentation in vitro, we elected to use the L929 fibroblast cell line, which was recommended by the company Aurogene Srl, Rome, Italy, as an ideal surrogate model to warrant efficiency and stability of transfection with PF62-pLentiPtpn22-R619W plasmid. Nevertheless, although ideal as a ‘surrogate’ simple model to explore the efficiency of lipoplexes in blocking the variant expression, this non-immune-derived cell line is not however ideal for exploring the impact of lipoplexes on downstream signaling pathways (such as TCR signaling, IL-2 secretion). We believe that this could be at best functionally evaluated in the natural context in vivo on PBMC immunocytes of the animal model carrying the R619W-Ptpn22 variant (see also reply to Reviewer 1 point 3 Nevertheless, we have already demonstrated in our previous publications the effects of the downstream effect on IL-2 secretion of lipolexes blocking human variant on human PBMCs (Refs. 17 and 19) of the revised marked version of the manuscript.

3) The qRT-PCR data are summarized but the manuscript does not specify biological replicates, variation, or normalization details. So, numerical data (mean ± SD) and complete statistical outcomes (F, df, p-values) are needed as well as confirmation that experiments were independently repeated.

The reported data were obtained from three independent biological replicates, derived from separate cultures of the same cell line as detailed on page 10, line 343, section 4.5.2 of Materials and Methods of the revised marked version of the manuscript.
Biological replicates were then used to perform two technical replicates of Real-Time PCR (qRT-PCR) in two independent experiments, in order to verify the reproducibility of the results (see also reply to Reviewer 1 point 2).

Expression levels were normalized to the housekeeping gene GAPDH, and relative expression was calculated using the ΔΔCt method, as detailed on page 10, line 362 of the revised marked version of the manuscript.

In this revised marked version of the manuscript, data are presented as mean ± SD, and the degrees of freedom (DFn, DFd) and F-values are provided in Supplementary Table 1 (see also additions within page 4 line 146; page 5 from line 149; page 10 from line 374).

4) The text refers to “previously established protocols” but omits characterization of the actual lipoplex batch used in this study. Were zeta potential, hydrodynamic diameter, and siRNA entrapment quantified for this experiment? Please include these parameters or a supplementary figure confirming reproducibility.

We thank the Reviewer for this inquire. We recently reported, in our previous study (Ref. 22 of the revised marked version of the manuscript), the structural characterization of the lipoplex prepared by incubating human siRNA with the same liposomal formulation [DOTAP/DOPE/DSPEPEG2000-Maleimeide (47.5/47.5/5 molar ratio)]. Here, we used the same liposomal formulation to deliver murine siRNA, which has the same net charge of the human variant and the same number of base pairs. As expected, we did not observe any difference between the size and more in general in the properties of the two lipoplexes differing in the siRNA variant. For this reason, we did not report structural data for lipoplex. However, we noticed that our previous sentence on page 7, row 233-234 was misleading. We changed it and explained this point more clearly in the revised version of the manuscript (see also the reply to Reviewer 1, point 4).

5) Figures 1–2 describe dose-dependent silencing, but quantitative knock-down percentages and ICâ‚…â‚€ values are missing. Please consider to include actual silencing efficiencies at each dose and time point.

We thank the reviewer for this observation. Quantitative knock-down percentages at each dose and time point are provided in Supplementary Figure 1 and on page 4, line 146; page 5, from line 149; page 10, from line 374 (see also reply to point 3 of this reviewer) of the revised marked version of the manuscript.

In our experiments, no lipoplex cytotoxicity was detected as assessed by monitoring cell morphology and viability, evaluating the quantity and quality of cell pellets, and measuring total protein concentrations at the end of the experimental procedure. Therefore, ICâ‚…â‚€ values, reported as mean ± SD, were included in the table below:

6) The absence of scrambled or non-targeting siRNA controls makes it difficult to exclude off-target or lipoplex-related effects. So, controls were included in this approach? if not, justify or include them in revised experiments.

We thank the reviewer for this important comment. In previous experiments, we already evaluated the use of an universal negative siRNA control (Sigma-Aldrich) formulated with liposomal formulations. That control showed no inhibitory or off-target effects on cell viability, morphology, or PTPN22 expression levels (Supplementary Figure 1 in Ref 17 of the revised marked version of the manuscript).

7) Lipofectamine. No mechanistic discussion of the observed lower efficiency is provided. Can the authors elaborate on the differences in uptake mechanism, stability, or endosomal escape that may explain the lower silencing efficacy of lipoplexes?

We thank the reviewer for this comment. The lower silencing efficiency observed with lipoplexes compared to Lipofectamine can be attributed to several factors related to cellular uptake and intracellular processing mechanisms. Lipofectamine-based systems are optimized commercial reagents that promote efficient siRNA complexation, membrane fusion, and endosomal escape, resulting in higher cytoplasmic release of siRNA molecules.

In contrast, the lipoplexes formulation used in this study may exhibit lower stability and less efficient endosomal release. Moreover, differences in lipid composition and charge ratio can influence the kinetics of complex disassembly and siRNA release within the cytoplasm (Ref. 28 of the revised marked version of the manuscript).8) ANOVA with Šidák’s test is mentioned, yet corresponding statistics are not reported (p-values, CIs) 

We thank the reviewer for this comment. The corresponding statistics are now reported at page 10, from line 374 of the revised marked version of the manuscript.

9) Considering the next step to in vivo experimental approaches. What endpoints (biodistribution, immunogenicity, toxicity) are planned for the NOD model, and what are the success criteria for moving toward clinical translation?

The protocol for animal experimentation aiming to assess biodistribution, efficacy and exclude immunogenicity and toxicity before undertaking clinical experimentation are reported in the Conclusion section on page 10, from line 387 of the revised marked version of the manuscript (see also reply to Reviewer 1 point 5).

10) There are some phrases that could be improved. Consider to proceed to a thorough grammar check to improve readability.

The manuscript underwent linguistic revision by a native English speaker as requested by both Reviewers

Round 2

Reviewer 1 Report

Comments and Suggestions for Authors

Although the manuscript has been carefully revised, the repetition rate is still at a relatively high level.

Author Response

Although the manuscript has been carefully revised, the repetition rate is still at a relatively high level.

We tried to eliminate as much as possible the overlapping throughout the manuscript detected by the program that probably relies on AI.

Nevertheless we noticed that the highest rate of overlapping was with the preprint of our publication in technical parts that could not be extensively modified.

Reviewer 2 Report

Comments and Suggestions for Authors

The authors addressed the comments raised 

Comments on the Quality of English Language

n/a

Author Response

See reply to Reviewer 1